# Ketogenic Diet for Cancer: Critical Assessment and Research Recommendations

**DOI:** 10.3390/nu13103562

**Published:** 2021-10-12

**Authors:** Jordin Lane, Nashira I. Brown, Shanquela Williams, Eric P. Plaisance, Kevin R. Fontaine

**Affiliations:** 1Department of Health Behavior, School of Public Health, University of Alabama, Birmingham, AL 35294, USA; jalane@uab.edu (J.L.); Nashirab@uab.edu (N.I.B.); iamshan@uab.edu (S.W.); 2Department of Human Studies, School of Education, University of Alabama, Birmingham, AL 35294, USA; plaisep@uab.edu

**Keywords:** ketogenic, cancer, adjuvant therapy

## Abstract

Despite remarkable improvements in screening, diagnosis, and targeted therapies, cancer remains the second leading cause of death in the United States. It is increasingly clear that diet and lifestyle practices play a substantial role in cancer development and progression. As such, various dietary compositions have been proposed for reducing cancer risk and as potential adjuvant therapies. In this article, we critically assess the preclinical and human trials on the effects of the ketogenic diet (KD, i.e., high-fat, moderate-to-low protein, and very-low carbohydrate content) for cancer-related outcomes. The mechanisms underlying the hypothesized effects of KD, most notably the Warburg Effect, suggest that restricting carbohydrate content may impede cancer development and progression via several pathways (e.g., tumor metabolism, gene expression). Overall, although preclinical studies suggest that KD has antitumor effects, prolongs survival, and prevents cancer development, human clinical trials are equivocal. Because of the lack of high-quality clinical trials, the effects of KD on cancer and as an adjunctive therapy are essentially unknown. We propose a set of research recommendations for clinical studies examining the effects of KD on cancer development and progression.

## 1. Introduction

Despite continued advances in screening, early diagnosis, and treatment, cancer remains the most dreaded of human maladies [1]. Surpassed only by heart disease as the leading cause of death in the United States, it is estimated that there will be 1,898,160 new cases and 608,570 deaths in 2021. The most common cancer sites are prostate, lung and colorectal for men; and breast, lung, and colorectal for women [2]. Lung cancer is the leading cause of cancer deaths for both sexes and is projected to remain so until 2040 [3], and likely well beyond.

Although tobacco remains the primary contributing factor for cancer development, other environmental factors, such as diet and lifestyle, play an extensive role. In 2015, it was estimated that diet accounts for approximately 30% of the attributable risk for cancer [4,5]. In 2017, the CDC estimated that 40% of all cancers are related to overweight and obesity (55% in women and 24% in men), with at least 13 different types of cancer linked to obesity (the most strongly linked were liver, endometrial, esophageal, and kidney) [6]. Although it is well-established that obesity associates strongly with both cancer incidence and mortality, it is less clear whether adiposity itself is the cause of or marker (byproduct) of underlying metabolic dysregulation that creates the conditions where cancer can develop and thrive [7]. The prevailing view has long been that positive energy balance resulting from excess energy consumption, lower energy expenditure, or both contribute to excess adiposity and subsequent manifestations of chronic disease, including cancer. However, emerging evidence suggests that dietary macronutrient composition may play a more extensive role than excess adiposity, per se in the development of cancer [8,9,10,11,12]. For example, high carbohydrate diets that are highly processed with added sugar have been shown to produce a hormonal milieu and metabolic derangements which promote the development of cancer and other chronic diseases. By extension, one might ask whether regulation of the quantity and/or quality of carbohydrates might mitigate cancer risk and/or cancer-related outcomes in those who develop cancer.

While several diets (e.g., vegan, Mediterranean) and dietary regimens (e.g., caloric restriction, intermittent fasting [13]) have been proposed as strategies for cancer prevention and as adjuvant therapies to standard-of-care cancer treatments, we provide a theoretical framework and preliminary evidence from preclinical and clinical studies on how the ketogenic diet (KD) may provide benefits in the prevention and treatment of cancer. Because there are several recent narrative, systematic, and meta-analytic reviews of KD for cancer [14,15,16,17,18], we focus on critically evaluating the state of the knowledge and provide a set of research recommendations to enhance the rigor and replicability of KD–cancer clinical applications and randomized clinical trials.

## 2. Ketosis and Spectrum of Ketogenic Diets (KD)

Nutritional ketosis has been defined as “the intentional restriction of dietary carbohydrate intake to accelerate the production of ketones and to induce a metabolic effect that stabilizes blood sugar, minimizes insulin release, and thereby mitigates the downstream anabolic and tumorigenic effects of longstanding insulin resistance [19] (p. 99).” Because maintaining stable blood glucose levels is essential for survival, even in the context of severe carbohydrate restriction, glucose can be synthesized from non-glucose substrates (e.g., certain amino acids) by hepatic gluconeogenesis (GNG). As part of a strategy to reduce the deleterious consequences and potentially lethal effects of unregulated protein depletion, mammals (including humans) evolved an efficient method to store excess energy. In a period of excess energy consumption, triglycerides consumed in the diet and produced from glucose and/or glucose in liver are transported to adipose where they are mobilized during prolonged fasting or starvation. Fatty acids released from triglycerides in adipose tissue are then transported to liver where they enter mitochondria and are partially diverted for ketone production—a primary source of energy in the brain during starvation as free fatty acids are unable to cross the blood–brain barrier and thus provide only a small amount of energy.

The classic KD is characterized by high-fat, moderate-to-low protein, and very-low carbohydrate content [20]. This translates into a dietary composition of about 90% fat, 2% carbohydrate, and 8% protein. As implied above, KD received its name because this diet induces physiologic ketosis which is manifested by increased concentrations of ketone bodies and decreased glucose and insulin concentrations in blood [21]. KD’s beneficial effects have been observed in a range of conditions including epilepsy and other neurologic diseases, obesity, type 2 diabetes, polycystic ovary syndrome, and cardiovascular disease (see [22,23] for a recent review).

The classic ketogenic diet consists of a ratio between fats and non-fats (carbohydrates + proteins) of 3:1 or 4:1. The major variations include: (1) Very Low-Calorie Ketogenic Diet is time-limited (~12 weeks) calorically restrictive (600–800 kcal), characterized by a minimum protein content (≥75 g/day), limited carbohydrate content (30–50 g/day), and a fixed amount of fat (20 g/day, mainly from olive oil and omega-3 fatty acids); and (2) the Low Glycemic Index Diet characterized by intake of a higher quantity of carbohydrates (60–80 g/day) from low glycemic index sources (e.g., lentils, chickpeas, bran cereals, carrots). Although not, strictly speaking, a KD, the Low Glycemic Index Diet has been effective in treating some forms of epilepsy and headaches [24] (it is thought that this diet, with its less restrictive carbohydrate intake, is unlikely to have beneficial effects on cancer [25]).

## 3. KD as a Therapeutic for Cancer: Hypothesized Mechanisms

While it is beyond the scope of this work to provide a comprehensive review of the proposed biological mechanisms by which a KD might confer benefits as a cancer therapy, (see [7,14,26,27,28] for more detailed expositions), we provide a brief and highly simplified overview, with particular emphasis on the rationale for proposing the potential value of a KD.

Despite their rapid proliferation, cancer cells use no more oxygen than non-cancer cells for oxidative purposes. Instead, they use about 10 times more glucose and produce about 70 times the rate of lactic acid than do normal cells. In other words, even with ample oxygen available, most cancer types derive energy from anaerobic glycolysis [29]. The reason that the vast majority (about 80%) of all cancers shift from oxidative phosphorylation to glycolysis (i.e., the Warburg Effect [30]) is unknown although it is speculated that doing so must confer a survival advantage (perhaps the acidic environment imposed by lactic acid is well tolerated by cancer cells, promoting further growth and spread to other organs [31]). Because the shift to glycolysis is manifested at the onset of tumorigenesis, many consider it one of the hallmarks of cancer [32]. Indeed, the Warburg Effect indirectly contributed to PET imaging, as the scan measures glucose disposal by cells (cancer cells take up far more glucose than surrounding cells, allowing contrasts in imaging).

Other factors, so-called nutrient sensors (e.g., insulin, insulinlike growth factor (IGF-1), mammalian target of rapamycin (mTOR), AMP-activated protein kinase (AMPK)) operate in the Warburg Effect, with their pathways playing important and complimentary roles in cellular proliferation and cancer expression [33,34,35,36,37].

Other potential metabolic pathways proposed as to why KD may confer benefits include the possibility that severely restricting carbohydrate intake alters mitochondrial function, the regulation of gene expression, the production of reactive oxygen species, the amino acid metabolism of cancer cells, angiogenesis and the vascularization of the tumor environment [38,39].

In summary, the primary rationale for proposing a KD as prevention or for treatment of cancer is to deprive cancer cells of their primary energy source, glucose, thereby interrupting the elaborate processes of nutrient sensors and other factors that are activated by the presence of glucose and insulin and appear to play important roles in their development and proliferation.

## 4. Preclinical Studies of KD for Cancer

Some animal models of cancer suggest that KD might be an efficacious cancer therapy when used alone or as an adjuvant to conventional therapies [14]. Specifically, some studies report that KD delays tumor development, slows growth, and increases survival time (e.g., [40,41]). Another set of studies show that KD may make tumor cells more vulnerable to the combination of chemotherapy and radiation as well as enhance the effects of targeted therapy (i.e., PI3K inhibitors) in tumor models [42]. However, other studies report increased tumor growth in rat models of kidney cancer [43] and mouse models of BRAF V600E-positive melanoma [44].

Li and colleagues [45] recently conducted a meta-analysis of 17 published animal studies to estimate KD’s potential antitumor effects. They found that KD, alone or in combination with caloric restriction, significantly reduced both tumor weight (standard mean difference [SMD] −2.45, *p* = 0.027) and volume (SMD = −0.76, *p* = 0.012) as well as prolonging survival time (SMD = 1.76, *p* = 0.003). Additional analyses suggested that KD ratio of 4:1 (i.e., severe carbohydrate restriction) was associated with the greatest increase in survival time (see also, [14,46,47]). Finally, the authors found that KD’s efficacy varied as a function of several factors, prompting them to conclude, “In summary, the pre-clinical evidence pointed toward an overall antitumor effect of the KD in animal studies currently available with limited tumor types. The efficacy of KD on tumorigenesis appears to be influenced by several factors, including cancer type or subtype, genetic background, cell line and/or model system, composition of the KD and tumor-associated syndromes. Therefore, more preclinical studies should be performed to elaborate the antitumor effect of KD in the future [45] (p. 11).”

## 5. Clinical Studies of KD and Cancer

Despite the promising results of KD from preclinical studies, there have been few human trials to isolate the effects of KD on cancer-related outcomes (most have focused on tolerability and safety [48]). For example, in a 4-week pilot study Fine et al. evaluated the safety and feasibility of a KD in 10 patients with different cancers [27]. Among the patients whose disease remained stable or partially remitted, they found ketone levels (i.e., serum beta-hydroxybutyrate [βHB]) on average, that were threefold higher compared with those with progressive disease. To date, most applications of KD in human cancers has been as an adjunctive therapy in conjunction with standard of care (i.e., chemotherapy, radiotherapy, and/or surgery). Recent evaluations of the literature conducted by Weber and associates (29 trials) [14], Talib et al. (14 trials) [48] and Yang and colleagues (6 trials) [15] Sremanakova and associates [18] (11 trials), Plotti et al. [49] (4 trials), and Romer and associates [16] (45 trials) among patients, virtually all being adults (i.e., 18 years of age and older), with a variety of cancers (e.g., glioblastoma, glioblastoma and gliomatosis cerebri, breast cancer, liver, pancreato-biliary cancer, lung and pancreatic, head and neck, colorectal cancer, and mixed cancer sites reported a wide range of favorable outcomes including progression-free survival, increased survival rate, increased rates of response to conventional treatment (i.e., stable disease after 6-week diet) [49], and enhanced quality of life (please see Table 1 for summary of clinical trials). While safe and well-tolerated by the majority of patients, some report side effects, including nausea, constipation, vomiting, hypoglycemia, and fatigue that may compromise adherence to KD [13,20]. Overall, while is has been found that KD may be beneficial for varying types of cancers as it relates to tumor characteristics, survival and side effects [50], it is important to underscore that, as described below, the trials were of varying methodological quality, which inhibits our ability to draw definitive conclusions on the effects of KD as an adjunctive therapy.

## 6. Limitations of Current Literature

Overall, the clinical trial literature on the use of KD as an adjunctive cancer therapy in humans has several important limitations that severely undermines our ability to make causal inferences concerning the effects of KD on cancer. The common theme of the limitations revolve around heterogeneity. That is, dramatic variations, within and between trials on many characteristics such as cancer type, time since diagnosis, patient characteristics (e.g., age, sex, overall health) KD variations, trial duration, study design, and outcomes assessment makes it impossible to draw conclusions on the effects on KD. In a sense, having so much variation in the published trials is a worse state-of-affairs than simply having an absence of trials because of the challenge in trying to draw conclusions from inconsistent findings, at least partly driven by the vast heterogeneity and varying methodological quality. Indeed, because of the vast heterogeneity of the human clinical trial literature, the validity of the published systematic reviews and meta-analytic reviews is highly questionable. For this reason, although preclinical evidence suggests favorable effects of KD, the human trials, to date, are equivocal regarding potential beneficial effects of KD as an adjunctive therapy, let alone as an intervention to impede cancer growth or improve survival. 

## 7. Conclusions and Future Directions

Preclinical studies in multiple strains of mice and types of cancer provide extensive evidence that the KD decreases tumor growth, prolongs survival, and reverses the process of cancer cachexia [14]. Clinical studies in humans are much more limited and have largely focused on small pilot or case studies and few clinical trials (see Table 2 for a summary of the strength of evidence for pre-clinical and human studies). Because of the promising effects in preclinical rodent models and the limited number of rigorous human clinical trials, it is clear that studies are needed in preclinical models and humans to understand the molecular mechanisms of KD and other low-carbohydrate diets in multiple forms of cancer. The hypothesized benefit of any low carbohydrate or low glycemic index diet is that the removal of processed foods containing sugar, added sugar, and lowering of starch-based carbohydrates reduce the amount of insulin required to clear a meal in the postprandial state. Since humans spend over 2/3 of their time in a postprandial state, it is logical to move forward under the supposition that lowering insulin could serve as a strategy to reduce risk of and progression of cancer. Presumably, decreasing the presentation of glucose by dietary carbohydrate restriction at the cellular and the epigenetic programming resulting from elevated insulin concentrations would be expected to reduce tumorigenesis and progression of cancer. In addition, insulin rapidly activates protein synthesis by activating components of protein translation such as eukaryotic initiation and elongation factors along with increasing the cellular content of ribosomes to augment the capacity for protein synthesis.

Studies are also needed to examine the effects of KD in multiple forms of cancer to determine whether the diet provides synergistic or additive benefits as an adjuvant therapy. Based on the sparse data available, there is reason to predict that KD could serve as an adjuvant to reduce tumor formation and progression. In addition, it will be important to examine tolerability of the KD in different types of cancers and treatments. If certain forms of cancers and/or treatments reduce palatability to the point where compliance is lost, then studies will be severely limited in scope and inference as it relates to the interpretation of findings. Thus, it will be important that future studies clearly define and test different levels of carbohydrate on low carbohydrate diets to improve the likelihood of success and to properly evaluate the effects of these diets on cancer risk and progression. It should be considered that the few studies which have examined the effects of KD on some forms of cancer and cancer treatment have observed an attenuation of skeletal muscle loss. While the mechanisms for this response are not entirely clear, preclinical studies from our group and others suggest that the ketone, beta-hydroxybutyrate (βHB), inhibits histone deacetylases which have been shown to preserve muscle in aging rodents [92]. These findings suggest that KD may reduce cancer cachexia and potentially improve functional capacity and quality of life while undergoing treatment.

Finally, with the commercial availability of exogenous ketone supplements, future studies are also needed to examine whether these supplements decrease cancer risk or progression. Little is known about the long-term effects of exogenous ketones in humans, but ketone esters and salt supplements transiently raise serum ketones, providing utility as a potential adjuvant treatment. Non-published observations from our group demonstrate that ketones consumed at or near the postprandial period reduce circulating levels of ketones and presumably have little effect on circulating insulin concentrations. Therefore, innovative dietary strategies with, perhaps, KD with ketone supplementation may be a favored strategy to increase circulating ketones while reducing insulin concentrations. Studies are needed to determine whether ketone supplements alone are sufficient, and at what dose and timing, to improve cancer and cancer-related outcomes.

## 8. Research Recommendations for Moving the Field Forward

Despite the metabolic rationale and relatively promising results in animal models, human trials testing KD as an adjunctive cancer therapy have been equivocal, indicating that we have a long way to go before drawing conclusions about the value of this diet. As noted above, the few human trials conducted thus far are fraught with methodological limitations, including, but not limited to, small sample sizes of heterogeneous patients (e.g., different cancer sites, disease durations, age, sex, comorbidities, among others), the absence of randomization and control groups, use of different and poorly described KD protocols, poor assessments of dietary adherence, short durations, and poorly defined and measured outcomes. The lack of high-quality trials, therefore, impedes both our scientific understanding and efforts to begin to translate a KD intervention into clinical practice. Without efforts to resolve these methodological limitations, the potential effects of KD on any cancer-related variables or outcomes will remain unknown. As noted by Romer and associates, “To form a final judgment about the efficiency of a KD in Oncology, a randomized controlled trial with a well-designed control group and sufficient power to also detect evidence for absence of antitumor effects is necessary [16] (p. 33).” Of course, not only would high-quality trials be required to detect potential antitumor effects but also on other important variables such as body composition, circulating insulin and inflammatory marker concentrations, side effects, functional capacity, survival time, and quality of life.

As such, we suggest that the following research recommendations may be useful in moving us toward a greater understanding of the effects of KD on cancer and related outcomes in humans (see also Figure 1).

Conduct small, rigorous non-randomized trials with homogeneous patient groups and common cancer sites to assess whether KD produces a “signal” on selected outcomes (particularly those related to response to standard care (e.g., effectiveness, side effects)) that would justify the conduct of larger, randomized-controlled trials.In randomized-controlled trials, provide sufficient detail of the KD and control diets (ensuring that they are comparable on vitamins, mineral and other nutrients) so they could be replicated by other investigators.Develop a standardized method to monitor and quantify adherence and tolerance to the KD (e.g., [93]).Develop a set of standardized assessments and outcome measures that include the full array of relevant variables (e.g., imaging of tumor characteristics, body composition, quality of life, and survival).Distinguish trials based on whether they attempt to isolate the unique effects of KD versus those which seek to estimate its effects as an adjunctive therapy.Examine the effects of exogenous ketones, alone and in conjunction with a KD, to determine whether they have synergistic or additive effects.Because it is unlikely that KD will cure cancer, trials should focus on whether KD reduces cancer progression or recurrence in those who experience remission through standard care.

Although outside of the scope of this paper, future studies should also address qualitative data and patient perceptions, such as quality of life assessments, that can be conducted alongside clinical trials.

Overall, the potential efficacy of KD for human cancers has yet to be determined. The vast heterogeneity of patients studied, in conjunction with the generally poor methodological quality of published trials has clouded our ability to estimate KD’s effects on the range of possible cancer-related outcomes. Until there is investment in providing adequate funding to conduct high-quality clinical trials, along with consensus and standardization around “best practices” among investigators, it is hard to see how our understanding of the effects of KD on cancer will advance.

## Figures and Tables

**Figure 1 nutrients-13-03562-f001:**
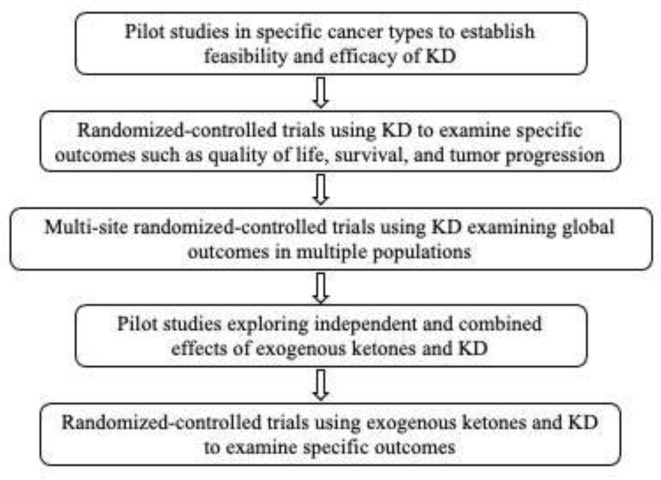
Sequential Research Recommendations for Investigating the Effects of the Ketogenic Diet (KD) on Human Cancers.

**Table 1 nutrients-13-03562-t001:** Summary of Clinical Studies of KD and Cancer.

Cancer Type(s)	Sample Size	Dietary Intervention	Study Duration	Results/Outcomes	References
Prostate	*N* = 45 Arm A: *N* = 27 Arm B: *N* = 18	Arm A: A low-carbohydrate diet, goal: (≤20 g per day), estimated actual carbohydrate intake: 37 g/day; Arm B: Control group (no dietary intervention)	6 months	-Weight loss -BMI reduction -Waist circumference reduction	[51]
Breast cancer	*N* = 60 Arm A: *N* = 30 Arm B: *N* = 30	Arm A: Medium-chain triglycerides (MCT) based ketogenic diet (6% calories from Carbohydrates [CHO], 19% protein, 20% MCT, 55% fat); Patients received 500 mL of MCT oil from the Nutricia Company every 2 weeks Arm B: Standard Diet (55% CHO, 15% protein, and 30% fat)	3 months	-Weight loss -BMI reduction -Reduction in body fat	[52]
Ovarian/endometrial cancer	*N* = 45 Arm A: *N* = 25 Arm B: *N* = 20	Arm A: Ketogenic diet (70% (≥125 g): 25% (≤100 g): 5% (<20 g) energy per day from fat, protein, and carbohydrates) Arm B: American Cancer Society diet (ACS: high in fiber, low in fat) Individual diet advice from certified dietitians. Weekly emails or phone calls. One face-to-face meeting after baseline assessment	3 months	-Self-reported improvement in energy levels (intervention group) -Fewer cravings for starchy foods and fast-food fats -Reduction in total body	[53,54]
Rectal cancer, head and neck cancer Breast cancer	*N* = 81 Arm A: *N* = 20 Arm B: *N* = 61	Arm A: ketogenic diet with additional consumption of non-glucogenic amino acids Arm B: no dietary intervention	30–40 days	-Decreased fat mass	[55]
Pancreatic cancer Duodenal cancer Common bile duct cancer Ampulla of Vater cancer Cholangiocarcinoma Neuroendocrine tumor	*N* = 19 Arm A: *N* = 10 Arm B: *N* = 9	Arm A: Ketogenic diet (3–6%, 14–27%; 70–80% energy per day from carbohydrates, protein, and fat) served as three meals and three snacks per day Arm B: usual Korean diet (55–65%, 7–20%, 15–30% energy per day from carbohydrates, protein and fat) served as three meals per day	12 days	-Decreased body cell mass higher in General Diet arm	[56]
Glioblastoma multiforme	*N* = 53 Arm A: *N* = 6 Arm B: *N* = 47	Arm A: self-administered KD Arm B: unspecified standard American diet	Duration: 3–12 months	- Two patients with grade 1 constipation, 4 patients with grade 1 fatigue, 1 patient with grade 2 fatigue, 1 patient with deep venous thrombosis during treatment, 1 patient with asymptomatic hypoglycemia, 1 patient with nephrolithiasis no grade 3 and higher toxicities or symptomatic hypoglycemia -Weight loss on non-calorie-restricted KD: 1 to 27 Ibs -Weight loss on calorie-restricted KD: 46 Ibs	[57]
Fearon et al. [44] Ovarian, Lung, Gastric	*N* = 5	Crossover study: Nasogastric tube feeding: normal, balanced regimen on days 1–6 KD containing same total calorie and protein on days 7–13	13 days	-Increase in body weight	[58]
Diverse	Recruited patients *N* = 12 Analyzed patients *N* = 10	KD with targeted CHO intake below 5% of total energy intake, written menus and samples of CHO-restriction products were provided	28 days	-Five patients with grade 2 fatigue, 5 patients with grade 1 constipation, 1 patient with grade 1 leg cramps -Weight loss - Decreased caloric intake -Adherence: 5 of 12 patients completed all 28 days of the diet	[27]
Diverse	Analyzed patients *N* = 78 Arm A: *N* = 7 Arm B: *N* = 6 Arm C: *N* = 65	Arm A: full adoption of a non-specified KD, patients informed about a single company producing KD-related food Arm B: partial adoption of a non-specified KD, patients informed about a single company producing KD related food Arm C: patients who did not adopt a KD	Not specified	1. Reduction in TKTL 1 was associated with adopting a KD; 2. Correlation between improvement in cancer status category and full adoption of a KD (χ2 = 33.26; df = 4; *p* = 0.00001	[59]
Diverse	Analyzed patients *N* = 6	Self-administered KD (recommended CHO intake < 50 g/day) during the course of RT/RCT; patients received basic information on KD; counseling at least once per week	Patient-dependent from 32 to 73 days	-Decreased fat mass	[60]
Glioblastoma	Assessed for eligibility: *N* = 57 Randomized: *N* = 12 Arm A: *N* = 6 Arm B: *N* = 6 Retention at 12 weeks. *N* = 4 Arm A: *N* = 3 Arm B: *N* = 1	Arm A: MCTKD (75%; 15%; 10% of energy per day from fat, protein and carbohydrates, with 30% of fat from MCT nutritional products) Arm B: MKD (80%; 15%; 5% of energy per day from fat, protein and carbohydrates)	12 weeks	1. Arm A: Three patients retained for 3 months (drop-out = 50%) Arm B: One patient retained for 3 months (drop-out = 83%) 2. GHS at baseline: Arm A: patients who later withdrew: 72.2 ± 20.7; patients who retained: 75 ± 6.8 Arm B: patients who later withdrew: 70 ± 13.8; patients who retained: 80 ± 0 GHS: at week 6: Arm A: patients who withdrew at week 6: 41.7 ± 0; patients who retained: 66.7 ± 0 Arm B: patients who withdrew at week 6: 50 ± 0; patients who retained: 100 ± 0 3. Adverse events during the first 6 weeks: Arm A: diarrhea (*n* = 1, CTCAE grade 1), nausea (*n* = 1, CTCAE grade 1), vomiting (*n* = 1, CTCAE grade 2), dyspepsia (*n* = 1, CTCAE grade 1) Arm B: vomiting (*n* = 1, CTCAE grade 1), dry mouth (*n* = 1 MKD, CTCAE grade 1)	[61]
Glioblastoma	Enrolled: *N* = 6 Completed intervention: *N* = 4	MKD (70%: 3–5% (≤20 g) energy per day from fat and carbohydrates; protein consumption was not restricted	12 weeks	-Constipation in two patients, resolved with dietary modification	[62]
Glioblastoma	Included patients *N* = 20 Evaluable for efficiency *N* = 17	KD with CO intake < 60 g/day, additionally highly fermented yoghurt drinks and two different plant oils were provided to be consumed at will. No calorie restriction, patients were instructed to always eat to satiety	Until progression of the disease	-Three out of 20 patients discontinued the diet after 2–3 weeks without progression, due to reduced QoL - Body weight reduction -Diarrhea, constipation, hunger and/or demand for glucose were present in some patients during the diet	[63]
Diverse	Enrolled: *N* = 16 Completed intervention: *N* = 5	KD with CHO limited to 70 g per day and 20 g per meal Two oil–protein shakes consumed in the morning and in the afternoon	12 weeks	-11/16 Patients discontinued the diet - 3/11 were unable to adhere to the diet, -6/11 discontinued due to progressive disease -2/11 died from progressive disease - reported side effects included increase in appetite loss, constipation, diarrhea and fatigue during the diet - QoL was low at baseline and stayed relatively stable during the intervention; worsening of fatigue, pain, dyspnea and role function but emotional functioning and insomnia improved slightly	[64]
Diverse	Enrolled: *N* = 17 Drop-out before first analysis: *N* = 6 Completed intervention: *N* = 4	Modified Atkins Diet with 20 to 40 g of CHO and restricted consumption of high CHO foods no restrictions for calories, protein or fats	16 weeks	-13/17 patients discontinued the diet before 16 weeks -weight loss -Reported adverse effects included: hyperuricemia (*N* = 7), hyperlipidemia (*N* = 2), pedal edema (*N* = 2), anemia (*N* = 2), halitosis (*N* = 2), pruritus (*N* = 2), hypoglycemia (*N* = 2), hyperkalemia (*N* = 2), hypokalemia (*N* = 2), hypomagnesemia (*N* = 2), flulike symptoms/fatigue (*N* = 2)	[65]
Glioblastoma multiforme	Phase A: *N* = 9 Phase B: *N* = 8 Completed intervention *N* = 6	Phase A: Fluid KD with a 4:1 ratio (4 g fat versus 1 g protein plus carbohydrates, 90% energy from fat) Patients were allowed a snack with the same 4:1 diet ratio once a day Phase B: Solid-food KD (diet ratio 1.5–2.0:1) with MCT; (70% energy from fat with the consistency of an emulsion)	14 weeks	-6/9 patients included in phase A completed the 14 weeks KD - Reported adverse effects included: constipation (*n* = 7), nausea/vomiting (*n* = 2), hypercholesterolemia (*n* = 1), hypoglycemia (*n* = 1), low carnitine (*n* = 1) and diarrhea (*n* = 1). CTCAE grade 2: hallucinations (*n* = 1), allergic reaction (*n* = 1) and wound infection (*n* = 1)	[66]
Glioma	*N* = 29	MAD with a 0.8–1:1 ratio (0.8-1 g fat to 1 g carbohydrate plus protein Duration: 6 weeks	6 weeks	-28/29 patients completed the 6-week diet - Reported adverse events: Grade 2 constipation (n = 1), grade 1 fatigue and nausea were present in the patients -Decreased BMI for all patients	[67]
Lung	Enrolled patients: *N* = 7 Completed intervention: *N* = 2	KD with 90%; 8%; 2% of energy per day from fat, protein and carbohydrates. All meals prepared for the patients	42 days	-Weight loss - Reported adverse events included: constipation, diarrhea, nausea, vomiting and fatigue; hyperuricemia	[68]
Pancreas	*N* = 2	KD with 90%; 8%; 2% of energy per day from fat, protein and carbohydrates. All meals readily prepared for the patients	34 days	-1/2 patients completed the intervention 2. Reported adverse events included: Constipation, diarrhea, nausea and vomiting, 1 patient experienced dehydration -Weight loss	[68]
Desmoid tumor	*N* = 1	TPN consisting of 28 kcal fat/kg body weight/day, 1.5 g protein/kg body weight/day; 40 g glucose/day	Desmoid tumor	-Body weight increased	[69]
Glioma	*N* = 2	ERKD: with a 3:1 ratio of ingested nutrients (3 g fat versus 1 g protein plus carbohydrates) 20% restriction of calories per day	12 months	-Adherence: 1/2 patients completed the intervention -Reported headaches -Initial body weight decrease in both patients and remained stable afterward	[70]
Glioblastoma multiforme	*N* = 1	ERKD delivering 600 kcal per day, consisting of 42 g fat, 32 g protein and 10 g CHO per day	56 days	-Bodyweight decreased in the first 14 days of the diet - Grade 4 hyperuricemia reported, resulted in diet change to calorie restricted non-ketogenic diet	[71]
Rectal	*N* = 1	Paleolithic KD, nutrients consumed in a fat:protein ratio of 2:1 animal fat, red meats and organ meats were encouraged, root vegetables were allowed, all other foods were prohibited	24 months	-Decreased bodyweight -Initial decrease in volume after concomitant radiotherapy -Tumor volume remained stable but four hepatic metastases were detected at the end of the diet	[72]
Diverse	*N* = 12	Single 3 h infusion of glucose-based (GTPN) or a lipid-based TPN (LTPN) containing 4 mg glucose/kg/min or 2 mg lipid/kg/min, respectively	3 h	-No statistically significant stimulation or suppression of FDG uptake	[73]
Recurrent Breast	*N* = 1	Self-administered high doses of oral vitamin D3 (10,000 IU/day), and KD rich in oleic acid. Duration: 3 weeks	3 weeks	-Progesterone receptor status positivity increased -HER2 positivity decreased	[74]
Astrocytoma	*N* = 2	KD with 60%; 20%; 10%, 10% of energy per day from MCT oil, protein, carbohydrates and dietary fat plus additional supplements	8 weeks	-Dose uptake ratio tumor: decreased normal cortex decreased -Adherence: 100% patients were able to complete the dietary intervention	[75]
Esophagus Stomach Colon-rectum	*N* = 27 Arm A: *N* = 9 Arm B: *N* = 9 Arm C: *N* = 9	Arm A: glucose-based TPN (100% of the calorie from dextrose); Arm B: lipid-based TPN (80% of the calorie from fat, 20% from dextrose); Arm C: oral diet All diets were iso-caloric and isonitrogenous. Duration: 2 weeks	2 weeks	No statistically significant changes	[76]
Head and neck	*N* = 12	Unspecified Western diet followed by unspecified KD	Variable, up to 4 days	Decline of mean lactate concentration in the tumor tissue during the KD	[77]
Brain	Included: *N* = 9 intervention: *N* = 5 retrospectively added control *N* = 4	KD based on ready-made formula, with a 4:1 ratio of ingested nutrients (4 g fat versus 1 g protein plus carbohydrates)	variable from 2 to 31 months	-Diet tolerated by 4/5 patients,(strict adherence only in 2 patients) -Four out of 50 MRI spectroscopy scans detected ketone bodies in the brains of the patients following the KD	[78]
Lung	*N* = 44	Mild KD (patients were encouraged to avoid high CHO food) in combination with HBO, hyperthermia and polychemotherapy administered during induced hypoglycemia	24 weeks	-Adverse events reported—during treatment period: grade 5 neutropenia (N = 1), grade 3 neutropenia (N = 3), grade 3 anemia (N = 10), grade 4 thrombocytopenia (N = 3), grade 3 fatigue (N = 5), grade 3 diarrhea (N = 8), grade 3 neuropathy (N = 1), all of which were attributed to chemotherapy	[79]
Pancreas	*N* = 25	Mild KD (patients were encouraged to avoid high CHO food) in combination with HBO, hyperthermia and polychemotherapy administered during induced hypoglycemia	Duration: mean follow-up: 25 months	-Adverse events reported: during treatment period: grade 3/4 neutropenia (N = 9), febrile neutropenia (N = 1), grade 3 anemia (N = 7), grade 4 thrombocytopenia (N = 4), grade 3 diarrhea (N = 2), all of which were attributed to chemotherapy	[80]
Brain	*N* = 8	MAD with20g CHO/day restriction	2-24 months: mean- 13 months	-7/8 completed intervention -Decreased body weight -Reduction in seizure frequency per week	[81]
Glioblastoma multiforme	*N* = 1	Energy-restricted KD with a 4:1 ratio of calorie intake (fat versus protein plus carbohydrates) Total calories calculated 25% below BMR	4 months	-No metabolically active tumor detected	[82]
Glioblastoma multiforme	*N* = 1	KD with a 4:1 ratio of calorie intake (fat versus protein plus carbohydrates), delivered as calorie-restricted diet, combined with intermittent fasting, HBOT, other novel therapies and SOC treatment	20 months	-Good surgical outcome and regressive changes in histopathology -Decreased body weight	[83]
Diverse	*N* = 6	Very low CHO diet (not further specified) with a multitude of supplements, including amino acids and Vitamin D^3^ combined with SOC therapy	Varied	-Shrinkage of tumor or stable disease was reported during the intervention -Subjective improvement reported in some cases	[84]
Head and neck	*N* = 14	KD with as little CHO as possible (estimated < 50 g per day), combined with insulin administration 3 × per day	Not specified	Visible remission after 2–3 weeks, but rebound effect after 2–3 months on the diet	[85]
Extra-cranial	*N* = 30	KD with as little CHO as possible (estimated < 50 g per day), combined with insulin administration 3 × per day	Not specified	Tumor shrinkage in some cases Improvement in general condition and positive effects on clinical symptoms	[86]
Exra-cranial	*N* = 23	KD with as little CHO as possible (estimated < 50 g per day), combined with insulin administration 3 × per day	Not specified	-Reduced pain severity, fatigue but deteriorated orientation	[87]
Pancreatic cancer Duodenal cancer Common bile duct cancer Ampulla of Vater cancer Neuroendocrine tumor	*N* = 18	LCKD: Energy content: 1500 kcal/d, provided 4% from carbohydrate, 16% from protein and 80% from fat. Ketogenic ratio of 1.75:1 (F: C + P w/w)	4 weeks	-Patients were in a poorer nutrition state after surgery, but this was alleviated at week 4; - LCKD induced ketone body production -Week 4, there were no significant differences in ketone levels	[88]
Glioma	*N* = 13 *newly diagnosed= 6* *recurrent=7*	KD + MCT + Metformin 850	6 weeks (recurrent) 2 weeks (newly diagnosed)	Increase in survival rate. Synergistic interaction between radiation therapy and KD.	[89]
Invasive Rectal	*N* = 359	KD ≥ 40% kcal fat and <100 g/day glycemic load (48)	Not specified	Reduced risk of cancer-specific deaths	[90]
Glioblastoma	*N* = 32	KD 50% kcal fat, 25% kcal CHO, 1.5 g/kg protein (17), CD (15)	3 months	No change in glucose increased ketosis No change in body weight	[91]

**Table 2 nutrients-13-03562-t002:** Overview of Strength of Evidence for Beneficial Effects of the Ketogenic Diet for Cancer and Related Outcomes in Pre-Clinical and Clinical Studies.

	Strength of Evidence
Strong	Moderate	Weak	Unknown
**Pre-Clinical Studies**
Tumor weight		X		
Antitumor effect/Tumor growth	X			
Progression-free survival				X
Tumor volume		X		
Overall survival time		X		
Cells’ responsiveness to therapy		X		
Body composition		X		
**Clinical Studies as an Adjunctive Therapy**
Tumor weight				X
Antitumor effect/Tumor growth				X
Progression-free survival				X
Tumor volume				X
Overall survival time				X
Cells’ responsiveness to therapy				X
Quality of life			X	
Body composition		X

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
