# Peer review of "Ketogenic Diet for Cancer: Critical Assessment and Research Recommendations"

_nutrients, 2021, doi:10.3390/nu13103562_

Round 1

Reviewer 1 Report

The manuscript depicts the state of the art of research about tumors and KD treatment. The article is well written and the critical assessment is clearly presented.

I would suggest the following minor revisions:

-The paragraph 'clinical studies of KD and cancer' would benefit of few more specific referrals to literature and available data. For instance, the paucity of data in pediatric age should be mentioned. Moreover, a graphic support/ table of available studies in literature might support the content explored in this paragraph.

-For research recommendations, it is worth to mention the importance of quality of life assessment (e.g. with standardized questionnaires) in these patients , to be conducted in parallel to other clinical evaluations.

Reviewer 2 Report

This is an interesting review, and the paper is generally well written. However, in my opinion the  manuscript has some shortcomings in regards to some data analyses and text, and I feel this unique dataset has not been utilized to its full extent.

Review articles are an attempt to sum up the current state of the research on a particular topic. The authors did not an extensive literature survey and searches for everything relevant to the topic of the present review, and did not sorted it all out into a coherent view of the ‘state-of-the-art’.

The present review manuscript does not describe the recent major advances and discoveries in a particular area of research, significant gaps in the research, and current debates and ideas of where research might go next.

Lack of the necessary detail for readers to fully understand. An extensive literature survey must to be done on the subject. Difficult to follow logic and poorly presented data.

Inaccurate conclusions on assumptions that are not supported by data.

Inappropriate reference citations ignoring the journals’ format.

Round 2

Reviewer 2 Report

Please find attached some comments related to the revised manuscript. 

The "improvements" are not comprehensive, i would rather call it "slim" and the authors argues not performing some state-of-the-art literature research based on reviewers' recommendations i find it. 
